# Differences between gridded population data impact measures of geographic access to healthcare in sub-Saharan Africa

Fleur Hierink [1,2 ✉], Gianluca Boo [3,4], Peter M. Macharia [5,6], Paul O. Ouma[5], Pablo Timoner[1,2], Marc Levy [7], Kevin Tschirhart[7], Stefan Leyk [8], Nicholas Oliphant[9], Andrew J. Tatem [3] & Nicolas Ray [1,2]

## Abstract

**Background** Access to healthcare is imperative to health equity and well-being. Geographic access to healthcare can be modeled using spatial datasets on local context, together with the distribution of existing health facilities and populations. Several population datasets are currently available, but their impact on accessibility analyses is unknown. In this study, we model the geographic accessibility of public health facilities at 100-meter resolution in sub-Saharan Africa and evaluate six of the most popular gridded population datasets for their impact on coverage statistics at different administrative levels.

**Methods** Travel time to nearest health facilities was calculated by overlaying health facility coordinates on top of a friction raster accounting for roads, landcover, and physical barriers. We then intersected six different gridded population datasets with our travel time estimates to determine accessibility coverages within various travel time thresholds (i.e., 30, 60, 90, 120, 150, and 180-min).

**Results** Here we show that differences in accessibility coverage can exceed 70% at the sub-national level, based on a one-hour travel time threshold. The differences are most notable in large and sparsely populated administrative units and dramatically shape patterns of healthcare accessibility at national and sub-national levels.

**Conclusions** The results of this study show how valuable and critical a comparative analysis between population datasets is for the derivation of coverage statistics that inform local policies and monitor global targets. Large differences exist between the datasets and the results underscore an essential source of uncertainty in accessibility analyses that should be systematically assessed.

## Plain Language Summary

Knowing where people reside and what health services are accessible to them in a timely manner can make a difference in life-or-death situations. Geographic models that mimic the journey of patients can help understand where people cannot access healthcare and can provide valuable insights for policy and research. Population distribution data is essential for these models, as it determines the relative coverage provided by the existing health system. However, there are several datasets available on population distribution that vary widely. In this study, we quantify the impact of using six different population data sets to calculate healthcare coverage in sub-Saharan Africa. Our results show large continental, national, and subnational differences between the different gridded population datasets, which can strongly influence the uncertainty of healthcare accessibility models and thus the decisions based on them.

[1] GeoHealth group, Institute of Global Health, Faculty of Medicine, University of Geneva, Geneva, Switzerland. [2] Institute for Environmental Sciences, University of Geneva, Geneva, Switzerland. [3] WorldPop, School of Geography and Environmental Science, University of Southampton, Southampton, UK. [4] Small Arms Survey, The Graduate Institute, Geneva, Switzerland. [5] Population Health Unit, Kenya Medical Research Institute - Wellcome Trust Research Programme, Nairobi, Kenya. [6] Centre for Health Informatics, Computing and Statistics, Lancaster Medical School, Lancaster University, Lancaster, UK. [7] CIESIN, The Center for International Earth Science Information Network, Columbia University, Palisades, NY, USA. [8] Department of Geography, University of Colorado in Boulder, Boulder, CO, USA. [9] The Global Fund to Fight AIDS, Tuberculosis and Malaria, Geneva, Switzerland. ✉email: fleur.hierink@unige.ch

Geographic access to healthcare is essential to ensure universal health coverage, a key target of the United Nations Sustainable Development Goals (SDGs)[1]. While geographic access is only one of many factors, such as affordability, availability, and acceptability[2–4], that impacts access to healthcare, it is fundamental to the organization of a health system as it determines the spatial reach of health services in relation to the population[5,6]. Modeling geographic access to healthcare is necessary to identify gaps in health system coverage and to support targeted health system optimization and planning, such as placement of new facilities, deployment of community health workers, or mobile outreach[7,8]. The key components of a geographic accessibility analysis are the population needing access, the locations of health facilities, and data to help model connectivity and travel time (i.e., road networks, land cover, streams, elevation, and care-seeking specificities)[5,9]. Although data on each of these components is increasingly available, accurate, and current[10], there are persistent differences between regions, hampering accessibility analyzes in data-poor regions[11]. Global advancements in population modeling have enabled the research community to use several gridded population datasets[12–17] in combination with recent data on health facility location[18], opening new avenues for modeling geographic accessibility to healthcare in data-poor settings. It is not known to what extent the use of different population data in accessibility analyzes affects accessibility coverage (i.e., the proportion of the population that can access a health facility within a given travel time threshold) and thus the monitoring of indicators that underpin policy-making at the global, national, and subnational level. This study aims to shed light on the magnitude and variation of these effects and possible policy implications, by conducting the first comprehensive comparison of six of the most commonly used global gridded population datasets in a geographic accessibility model at 100-meter resolution for sub-Saharan Africa.

Gridded population datasets allocate population counts across rows and columns of grid cells either by using simple techniques to uniformly redistribute census data or by using ancillary variables derived from Earth observations (e.g., land cover, elevation, and night lights) or socio-economic data to apply dasymetric modeling techniques, that provide more refined population estimates[19]. These datasets typically use a country's most recent census or projected estimates, summarized in available administrative units or census enumeration areas, to disaggregate population numbers at a finer spatial and temporal resolution[20–22]. Population redistribution techniques vary from dataset to dataset, meaning that the suitability of each dataset for any spatial analysis is context-dependent. Discrepancies between datasets do not necessarily reflect specific appropriateness; rather the suitability of each gridded population dataset is highly dependent on the target scale, context and purpose, and geographic extent of the analysis[19]. However, even when two or more gridded population datasets meet some predetermined criteria, differences in accessibility coverage may be observed. Different population data have been used in accessibility analyzes, exposing potential uncertainty in accessibility coverage estimates and making comparability across studies difficult. Some studies have used national censuses[23], WorldPop products[7,11,24–28], Gridded Population of the World (GPWv4)[29], High-Resolution Settlement Layer (HRSL)[30], or LandScan[31]. The scientific literature increasingly acknowledges differences between gridded population datasets[19,20]. However, the focus is often on general data characteristics and their suitability[19,20,32] or on the country- or discipline-specific implications of using the different data products[21,33–35], rather than quantifying differences in model outcomes at large geographical scales. In addition, the motivation and implications of using a particular population dataset are usually neglected in accessibility studies[35,36]. The choice of any specific population layer is likely driven by personal preferences, lack of knowledge of other sources, or ease of access and use.

Here, we systematically assess differences between estimates of geographic healthcare accessibility for all of sub-Saharan Africa using the most popular gridded population data products: (1) WorldPop top–down constrained, (2) WorldPop top–down unconstrained, (3) HRSL, (4) GPWv4, (5) LandScan, and (6) Global Human Settlement Population (GHS-POP). Healthcare accessibility is modeled at 100-meter resolution using the most recent release of the geocoded health facility inventory of 50 countries in sub-Saharan Africa to enable a fair comparison between models[18]. We contrast accessibility coverage statistics derived from the six population datasets, across countries at national and subnational scale. Travel time was calculated by developing a friction layer at 100 meters resolution, representing the estimated time required to reach the nearest health facility. We intersected the six different gridded population datasets with our travel time estimates to determine accessibility coverages within various travel time thresholds (i.e., 30, 60, 90, 120, 150, and 180-min).

Our accessibility coverages vary widely between the different datasets and estimates on the sub-Saharan African level mask larger subnational variations. Differences are most pronounced in scarcely settled regions, where administrative units are large. Datasets that distribute population over larger land areas, rather than being limited to building footprints, result in longer travel times for a portion of the population and therefore lower overall estimates of accessibility, notably changing accessibility patterns. The results provide useful clues for policy-making and critical reflection on previous estimates of accessibility to healthcare and their associated uncertainties.

## Methods
In order to quantify and compare the differences in healthcare coverage between the six different datasets, we took several steps to prepare, process, and analyze the spatial data.

**Accessibility model.** Accessibility to healthcare was modeled in terms of travel time to the nearest public health facility. This calculation was made by overlaying health facility coordinates on top of a friction raster. Each grid cell in the friction raster represented a unique land cover class which was assigned a travel speed. On-road travel represented motorized speeds whereas for off-road travel walking speeds were used. The cumulative time required to traverse all cells to the nearest health facility was then calculated for each grid cell which represents the travel time raster. This calculation was done on the eight-directional least-cost path algorithm[9,37] and was isotropic, meaning that no corrections were made for slopes. Although anisotropic analyzes make the model results more realistic, we preferred an isotropic analysis to minimize model complexity and assumptions, in the absence of local transport information. Slope corrections are usually applied to the speeds of pedestrians and cyclists. It is therefore important to have local information on modes of transport, and this is likely to vary from country to country and from region to region.

The friction raster represents information about potential impacts on a patient's journey to healthcare, including land cover type, barriers to movement, and the road network. All this information was extracted from open data sources and processed between January 2021 and October 2021, however, reference dates of some of the data can date back up until 2015 as indicated in Table 1 and Supplementary Table 2. (Table 1). We fully automated the entire workflow in an R and Python environment (Supplementary Fig. 1). In brief, road networks, rivers, and lakes were extracted from OpenStreetMap (OSM) using the

**Table 1 Overview of spatial data sources used in the study.**

| Dataset | Producer | Resolution | Year | Citation |
|---|---|---|---|---|
| Landcover | Copernicus | ~100 meters | 2019 | 40 |
| Roads | OpenStreetMap | Vectorized | 2021 | 38,71 |
| Waterbodies (lines and polygons) | OpenStreetMap | Vectorized | 2021 | 38,71 |
| Health facilities | Maina et al. (2019) | Vectorized | 2018 | 18 |
| Travel scenario | Adapted from Weiss et al. (2020) | – | – | 36 |
| Administrative boundaries | Global Administrative Areas (GADM) | vectorized | 2020 | 41 |
| Mean administrative unit area for publicly available population census data | Center for International Earth Science Information Network - CIESIN | ~1 kilometer | 2018 | 53 |

For an overview of the different gridded population data products please see Supplementary Table 2.

*osmextract*[38] library in R[39] (version 4.0.4). The land cover for sub-Saharan Africa was downloaded at 100-meter resolution from Copernicus[40]. Health facility coordinates were extracted from a geocoded database for sub-Saharan Africa[18]. Administrative boundaries for all African countries were taken from the database of Global Administrative Areas (GADM)[41].

Data preparation was done on a per-country basis and optimized to minimize computation time as detailed in Supplementary Fig. 1, implying that land cover data was first downloaded for the entire African continent and then processed for each country, separately. In summary, and as shown in Supplementary Fig. 1, data processing included cropping to the bounding box of each country to minimize computation time in the masking step. Then rasters were clipped to exact country borders. Lastly, the land cover raster was projected in the country's coordinate system (Supplementary Table 1).

The process was parallelized using the *doParallel*[42] and *foreach*[43] R libraries. All necessary data processing steps were done using the *terra* package[44]. Scripts for data processing and analysis can be sourced from Github [https://github.com/fleurhierink/Population_Access] and Zenodo [https://doi.org/10.5281/zenodo.7004009][45].

Vector data representing road networks and barriers to movement were fetched using the *osmextract*[38] library in R[39] (version 4.0.4) and projected in the country's coordinate system (Supplementary Table 1). All road classes that are officially classified by OSM were included for analysis[46]. Barriers to movement (unless a road crosses over) included hydrographic lines classified as river and hydrographic polygons. Streams and smaller waterbodies were excluded from the analysis since they can be traversed with ease[46].

The geocoded inventory of public health facilities in sub-Saharan Africa[18] assembled between 2012 and 2018 was downloaded and projected to match the spatial coordinate system of the other datasets by country (Supplementary Table 1). We included all health facilities irrespective of type (e.g., primary, secondary, health centers, etc.).

Finally, all data were combined in a friction raster at 100-meter resolution. This resolution offered the best compromise between computational efficiency, spatial detail to address fine-scale disparities in healthcare access, and consistency with the assembled spatial data described above. The vector data were rasterized at 100-meter resolution. All raster cells were aligned, and layers merged to create one comprehensive land cover raster, to which travel scenarios (Supplementary Data 1) were applied. The travel scenarios for all sub-Saharan African countries were taken from Weiss et al. (2020)[36], but adapted to the context of this paper (Supplementary Data 1). When a travel scenario from Weiss et al.[36] did not indicate a speed for a specific road class in a given country, we used the African average travel speed for that road class (Supplementary Data 1).

We did not use an existing travel time surface, such as the one available from Weiss et al.[36], because its coarser resolution (i.e., 1 km × 1 km) did not match the resolution of most of the input- and population data, nor our objective to capture barriers with higher spatial accuracy. In addition, the assumptions made by Weiss et al.[36] about travel speeds and barriers to movement, such as the traversability of waterbodies at a speed of 1 km/h and the use of global average speeds for road classes for which no information on speed limits was available, did not fit well in the context of sub-Saharan Africa. Most importantly, the travel time surfaces modeled in this study were used as an indicator to assess the impact of using different gridded population products, rather than to inform the research community on coverage statistics. To inform policy- and decision-making, it is preferable to work at a finer spatial scale to ensure greater accuracy and robustness in the model inputs by consulting local experts on health facility data and information of health seeking behavior of the target population, so that the travel scenarios can be best adapted to the local context.

**Data processing of population grids.** In Supplementary Table 2, the properties of the different gridded population datasets are described. All population rasters were clipped to country borders and reprojected to each country's projection system (Supplementary Table 1). Population that was lost from the original files, due to these data processing steps, were equally smoothed out over the rasters so that total population counts remained the same as in the original files. This was done by comparing the summed population at administrative level 2 for the original and projected rasters. Due to the different resolutions of the datasets, and to avoid resampling of population raster data, all population grids were transformed into spatial points representing the centroids of the grid cells.

**Extraction of accessibility coverage statistics.** To assess the spatial variation in national and subnational accessibility coverage statistics, we overlaid the six gridded population datasets onto the travel time rasters for each country. We extracted the travel time and the administrative boundary (level 1 and 2) for each population point feature. We then calculated the accessibility coverage statistics, by means of zonal statistics, to output the population able to reach the nearest health facility within a certain travel time. Both relative and absolute coverage statistics were obtained per administrative unit. Population falling on barriers (i.e., waterbodies or just outside country borders) were not included in the extraction of coverage statistics. The absolute and relative number of people falling on barriers are indicated in Supplementary Data 2.

**Limitations of method.** We note that our travel time grid, which captures the accessibility of the nearest health facility, served as

the main input data for deriving the coverage statistics presented. However, we recognize that realistic estimates of geographic access to healthcare require local knowledge of health-seeking behavior, such as travel modes and speed, as well as information on (seasonal) barriers to mobility. Although we have used local expert knowledge to build accessibility models in previous studies[28,30,47], the scale and context of the present analysis did not allow us to use such local knowledge. Such detailed input was beyond the scope of this study, which aims to reflect important differences between population datasets. Therefore, our travel time maps and associated accessibility estimates should not be used for health system planning at national and subnational levels. However, our methodology can be adapted to local contexts, drawing on the expertize of different stakeholders at national and subnational levels, particularly in relation to transport modes and speeds.

A limitation of the current study is that the unconstrained datasets included a proportionally higher number of people living in areas considered to be barriers (i.e. waterbodies or areas outside national borders).

In addition, modeling geographic accessibility presents challenges other than differences between gridded population datasets. For example, uncertainties in travel modes and speeds can lead to under- or overestimation of accessibility. If travel speeds are assumed to be higher than they actually are, the accessibility model results will incorrectly indicate a higher accessibility coverage. This also applies to uncertainties in road network data when some roads maybe missing or when roads may actually present dirt tracks that in reality cannot be traveled by motorized vehicles. Realistic modeling of access to healthcare is therefore highly dependent on reliable and locally agreed model inputs. One nascent area is the use of Google Maps APIs to characterize travel time which has been shown to estimate near to reality travel times in urban areas. The approach potentially accounts for traffic, weather conditions, difference in speeds, road conditions and other predisposing factors. However, the approach is still at an early stage of development and is more applicable in urban areas where data collection through voluntary geographic research is better than in remote and rural areas where the majority of people live. Therefore, the use of least-cost path algorithms still remains feasible but requires improved parameterization[48].

**Reporting summary**. Further information on research design is available in the Nature Research Reporting Summary linked to this article.

## Results

**Diverging accessibility coverage estimates for sub-Saharan Africa**. Estimates of accessibility coverage, modeled by constructing a travel time grid at 100-meter resolution for all of sub-Saharan Africa (Supplementary Fig. 2), show greatly divergent results using the six different population datasets (Fig. 1a, b). Importantly, HRSL data for Ethiopia, Somalia, South Sudan, and Sudan were not available at the time of this study.

For all of sub-Saharan Africa, the population that has access to healthcare is highest when using HRSL, followed by GHS-POP (Fig. 1b). Differences in accessibility coverage are larger at 30- and 60-min catchments and logically decrease as travel times increase. An estimated 88.2% of the HRSL-derived population has access to a health facility within 30 min travel time. This value drops to 60.5% when GPWv4 is considered (Table 2 and Fig. 1b). Access to healthcare is in general substantially lower when statistics are derived using GPWv4 and WorldPop top–down unconstrained datasets (Fig. 1b). These two datasets also present

the largest differences in accessibility coverage as compared to the other datasets (Fig. 2). Although the differences between the other datasets are smaller, there are still coverage differences of up to 9.5% among the other population products at 30 min travel time (Fig. 2). The relative differences are smallest between LandScan and WorldPop top–down constrained and between HRSL and GPWv4. While accessibility coverages at the sub-Saharan African level already show strong variation, such continental summary statistics substantially mask even greater variations at the national and subnational level.

**The comparison of national differences in accessibility enables the localization of major differences**. Moving from sub-Sahara Africa to national coverage statistics, we find new patterns, with varying results across population datasets and between countries (Fig. 3 and Supplementary Table 3). Strongly divergent trends are particularly evident in some countries, including Chad, Sudan, Eritrea, South Sudan, Central African Republic, Republic of the Congo, Democratic Republic of the Congo, Equatorial Guinea, and Gabon (Fig. 3). In these countries, we observe lower coverage statistics for GPWv4 and WorldPop top–down unconstrained, meaning that a relatively smaller number of people have access to healthcare, sometimes followed by notable discrepancies between coverage values for the other datasets. The differences in accessibility coverage can exceed 60% and would affect any conclusion drawn from one of the individual population datasets. In the Republic of the Congo, for example, accessibility coverage at 30 min travel time ranges from 28.8% to 88.9%. Using GPWv4 or WorldPop top–down unconstrained suggests that 71.2% or 65.5% of the population in the country is unable to reach the nearest health facility within half an hour travel time. In contrast, using GHS-POP, HRSL, LandScan, or WorldPop top–down constrained indicates that 11.1%, 13.9%, 15.8%, or 27.3% of the population is unable to reach healthcare within half an hour. This discrepancy between the datasets may have a strong impact on the conclusions drawn from monitoring global and national indicators of access to healthcare, and thus on decision making for resource allocation.

**Differences in accessibility coverage are most evident at the subnational scale**. Figure 4 illustrates accessibility coverage within 1-hour catchments at the subnational level (i.e., administrative level 1). Supplementary Data 1 presents accessibility coverage for 30, 60, 90, 120, 150, and 180 min travel time at administrative level 1 and 2. While the definition of administrative levels varies from country to country, administrative level 0 always represents the national borders of a country, administrative level 1 represents the largest subnational unit of a country, smaller subnational levels are levels 2, 3, and 4.

Despite the similarities in overall accessibility patterns, with low access in northern and central sub-Saharan Africa and higher access in southern sub-Saharan Africa and coastal regions, subnational differences between the datasets are clearly evident. Low accessibility coverage is particularly widely spread for GPWv4 and WorldPop top–down unconstrained. In Fig. 5 we present the average percentage point difference between the datasets we observed at the subnational level. The average difference between all datasets can be as high as 45.4%. However, when comparing individual datasets, the subnational average difference can exceed 70% (Fig. 5b and Supplementary Figs. 3–51).

**Explaining discrepancies in coverage estimates**. Most of the observed discrepancies in accessibility coverage can be explained by the characteristics and quality of the input data and the

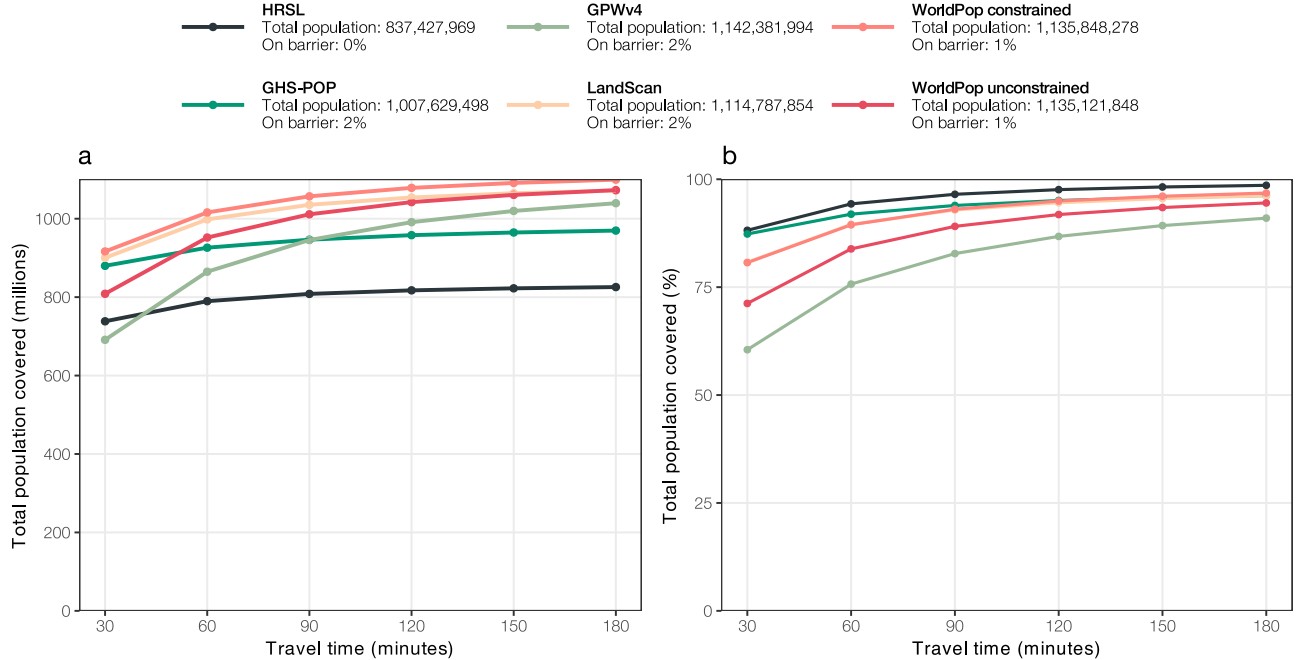

**Fig. 1 Accessibility coverage at the sub-Saharan African level.** Absolute (**a**) and relative (**b**) accessibility coverage for the six different gridded population data sets: HRSL, GHS-POP, GPWv4, LandScan, WorldPop top–down constrained, WorldPop top–down unconstrained. Total population is lower for HRSL because Ethiopia, Somalia, Sudan, and South Sudan are not included in the data set released in 2018. Legend indicates the total population falling on barriers (i.e., permanent waterbodies) and thus not included in the analysis.

redistribution approach used for creating the gridded population datasets. More specifically, the main differences in accessibility coverage that we observe can be explained by (1) the use of settlement data to conditionally constrain population to buildings, (2) the quality and resolution of the settlement data used, and (3) the granularity of the smallest publicly available unit for population data. In Figs. 4 and 5, the differences in accessibility coverage are particularly evident between datasets that constrain population to settlements (i.e., WorldPop top–down constrained, HRSL, GHS-POP, and LandScan) and the other datasets that allocate population based on proportional weighting or other areal interpolation techniques used for dasymetric refinement. Constrained population datasets typically use building footprints or settlement feature data derived from satellite imagery to constrain the distribution of population to grid cells in which buildings have been detected. The datasets based on settlement data have a large proportion of zero cells in areas where no buildings are detected[33]. This means that population is commonly distributed over smaller areas and therefore more concentrated in regions with human activity and health facilities. In contrast, datasets that do not contain information on settlements have a small proportion of zero cells. This is a natural consequence of using approaches that spread population over vast areas of land where few or no people are likely to reside, including extremely uninhabitable areas such as deserts or dense forests where there are no health facilities. These distorted distributions ultimately result in longer travel times for some of the population and therefore smaller overall accessibility estimates.

In the region of Borkou in northern Chad, for example, accessibility coverage is between 58.1% and 72.4% at 30 min travel time using HRSL, GHS-POP, LandScan, or WorldPop top–down constrained, and drops to almost 0% when GPWv4 or WorldPop top–down unconstrained is considered (Supplementary Figs. 6 and 45). Similar patterns were also observed in northern Niger and other regions south of the Sahara Desert. This region is sparsely populated and has large differences in

accessibility patterns between the datasets. Figure 6 shows an example of the observed visual differences between the datasets. The same is true for some regions in central sub-Saharan Africa, such as the Republic of the Congo, Gabon, and the Democratic Republic of the Congo where large areas of land are characterized by dense and closed forests with very few detected settlements (Supplementary Figs. 11, 12, 18 and 52). In Ogooué-Maritime, a province in western Gabon characterized by dense forests, accessibility coverage within 30 min ranged from 87.9% to 96.3% when using WorldPop top–down constrained, LandScan, HRSL, or GHS-POP, in ascending order of coverage. However, accessibility coverage decreases to 11.1% and 3.8% when WorldPop top–down unconstrained and GPWv4 are used (Supplementary Fig. 18). Comparisons of accessibility coverage between the settlement-based population data also show discrepancies (Figs. 2 and 5), as their accuracy appears to be highly dependent on the completeness of identified building structures. The quality of the underlying satellite data containing information on built environments and the applied methodology to automatically extract built features involves omission and commission errors, leading to an under- or overestimation of uninhabited areas[21,49,50]. While WorldPop top–down constrained uses polygon building footprint data and HRSL uses high resolution satellite imagery (~50 cm), GHS-POP extracts built features from Landsat 8 imagery with a resolution of ~30 meters[32]. Due to the difficulty of detecting built-up areas from coarser resolution satellite imagery, GHS-POP and, to a lower extent, LandScan have previously been found to overestimate uninhabited zones and thus underestimate people in sparsely populated sub-urban and rural areas[21,51,52]. We found similar patterns in two rural areas in Garissa and Nakuru counties in Kenya, where divergent patterns of settlement detection between the gridded population products were seen (Supplementary Figs. 53 and 54). Particularly GHS-POP did not seem to allocate population in small settlements that were included in the other datasets (Supplementary Fig. 54). When no population is

**Table 2 Summary coverage statistics for sub-Saharan Africa.**

| | Total population | 30 min | | 60 min | | 90 min | | 120 min | | 150 min | | 180 min | |
|---|---|---|---|---|---|---|---|---|---|---|---|---|---|
| | | nr. covered | % covered | nr. covered | % covered | nr. covered | % covered | nr. covered | % covered | nr. covered | % covered | nr. covered | % covered |
| HRSL | 837,427,969 | 738,362,867 | 88.2 | 789,665,384 | 94.3 | 808,260,473 | 96.5 | 817,316,977 | 97.6 | 822,468,235 | 98.2 | 825,661,368 | 98.6 |
| GHS-POP | 1,007,629,498 | 879,872,628 | 87.3 | 926,126,071 | 91.9 | 946,476,919 | 93.9 | 958,041,182 | 95.1 | 964,843,199 | 95.8 | 969,695,172 | 96.2 |
| GPWv4 | 1,142,381,994 | 691,184,991 | 60.5 | 864,838,798 | 75.7 | 945,653,297 | 82.8 | 991,147,422 | 86.8 | 1,019,755,993 | 89.3 | 1,039,511,968 | 91.0 |
| LandScan | 1,114,787,854 | 900,416,765 | 80.8 | 998,239,544 | 89.5 | 1,035,462,222 | 92.9 | 1,054,119,811 | 94.6 | 1,064,963,010 | 95.5 | 1,071,921,785 | 96.2 |
| WorldPop constrained | 1,135,848,278 | 916,456,304 | 80.7 | 1,015,980,775 | 89.4 | 1,057,068,975 | 93.1 | 1,078,609,302 | 95.0 | 1,091,225,178 | 96.1 | 1,099,582,164 | 96.8 |
| WorldPop unconstrained | 1,135,121,848 | 808,494,660 | 71.2 | 951,961,369 | 83.9 | 1,011,228,872 | 89.1 | 1,042,316,719 | 91.8 | 1,060,764,621 | 93.4 | 1,072,860,699 | 94.5 |

Absolute and relative accessibility coverage as visually presented in Fig. 1 for the six different gridded population data sets: HRSL, GHS-POP, GPWv4, LandScan, WorldPop top–down constrained, WorldPop top–down unconstrained. Total population is lower for HRSL because Ethiopia, Somalia, Sudan, and South Sudan are not included in the dataset released in 2018.

allocated to small rural settlements, a relatively large proportion of the population is distributed into larger built areas where facilities are located, this likely contributes to higher accessibility coverage statistics for GHS-POP and LandScan as compared to HRSL and WorldPop top–down constrained.

An important challenge for all gridded population datasets is the quality and granularity of the input population data. Even though census data is often collected at the household level or in smaller enumeration areas, countries usually release aggregated data at specific administrative levels to protect privacy[19]. The scale at which the latest population census is made publicly available[53] varies widely across sub-Saharan Africa (Fig. 7a) and ranges on average from ~2 km² to 182,211 km². Figure 7b illustrates the association between population input unit size (km²), relative coverage difference between the datasets at 1-hour travel time, and average total population per administrative unit (level 1). The figure shows that in areas where there are large differences in accessibility coverage between the datasets, the size of the population input unit is generally large, and the total population living in these units is small, mostly in the first or second quantile (Fig. 7b, top right corner). This means that when population counts in sparsely populated areas are aggregated into large units, differences between the datasets are greatest. Figures 5 and 7 show similarities between areas with high accessibility coverage differences and regions with large population input sizes, such as the northern- and central parts of sub-Saharan Africa. Sangha, for instance, a region in the Republic of the Congo has one of the highest average accessibility coverage differences across all datasets (45.4%) (Supplementary Figs. 7–55). The average total population of 45,281 people is spread out over ~57,686 km² land and the landscape is primarily comprised of dense forests, complicating building detection. The same is true for an area that we described before, Ougooué-Maritime province in Gabon, where the average coverage difference is 45%, the average total population is 44,230, the population input unit size is 7528 km², and the landscape is dominated by dense forests (Supplementary Fig. 2).

The aggregated nature of the input population data masks the spatial variability in population distribution at finer scales and therefore causes uncertainty when total population counts are reallocated into grid cells. Our analysis suggests that particularly in sparsely populated areas where population data is made available at a coarse scale, the different redistribution techniques used to create the different datasets cause most of the observed variation in the population reallocation patterns and thus translate into widely ranging accessibility coverage estimates.

## Discussion

Data on population distribution is the main denominator for almost all public health interventions. The effectiveness of evidence-based health planning, such as the distribution of health facilities or the implementation of vaccination campaigns, largely depends on accurate population estimates[54,55] to calculate resource needs and measure the impact of interventions[56,57]. Moreover, the SDGs and other international health targets are based on indicators that reflect the proportion of the population that has access to certain services. Knowing how many people live where is essential for these calculations[56]. Here we show that estimates of healthcare coverage vary widely depending on the gridded population dataset chosen and that they can lead to conflicting conclusions.

Our results show notable variations and tend to diverge most in regions with a low population density where administrative units are large, and land cover classes such as dense forests and deserts indicate sparse population distribution. The large

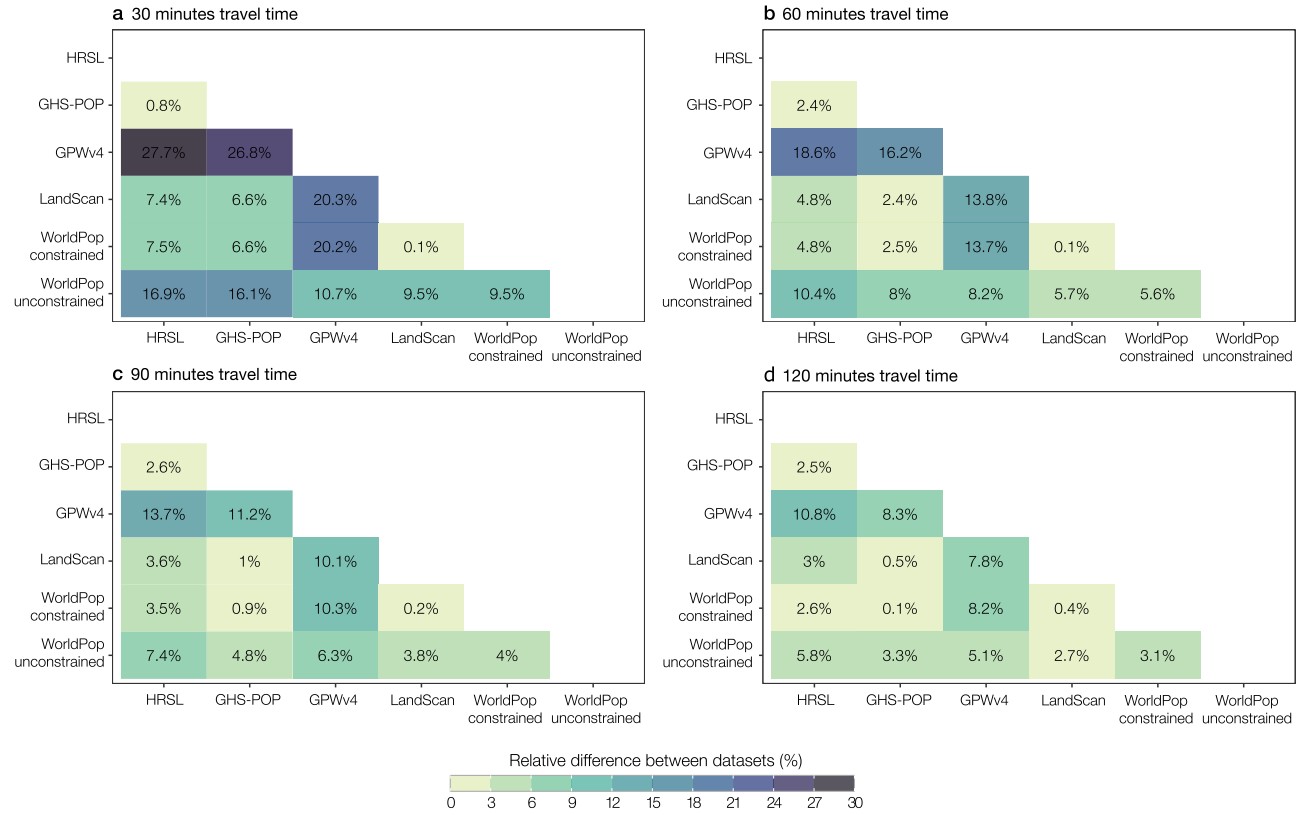

**Fig. 2 Relative difference in accessibility coverage estimates.** The matrix shows the relative difference in accessibility coverage statistics at **a** 30, **b** 60, **c** 90, and **d** 120 min travel time for the six gridded population datasets for the full Sub-Saharan African region.

variability in our results would also hold true for coverage estimates of other types of services for which similar accessibility models can be used, such as school access for children[58] within a predefined threshold or for estimates related to the people at risk of infectious diseases[59,60], people living in disaster-prone areas[33], or modeling vaccination coverage[61,62].

The use of one population dataset can have strong implications for policy- and decision-making. With new global targets aimed at improving access to healthcare, it is crucial that indicators that monitor progress are correct and based on realistic input parameter values. For instance, the recently adopted target 4 indicator of the World Health Organization (WHO)'s *Ending preventable maternal mortality* (EPMM) strategies states that by 2025 (1) at least 60% of the global population should be able to access the closest functional emergency obstetric care (EmOC) health facility within two hours travel time, and (2) 80% of countries should have a 2-hour accessibility coverage greater than 50%[63]. A United Nations (UN)-led guidance to help countries model this indicator will be released in 2022. In that context, our results can provide useful quantification of the expected relative differences and thus the sensitivity of this indicator based on various population datasets. Taking again the Republic of the Congo as an example, we find that when using different population datasets, accessibility coverage at 2-hour travel time ranges from 44.7% to 95.0%. Coverage statistics are highest when using GHS-POP (95.0%), HRSL (93.2%), LandScan (93.4%), or WorldPop top–down constrained (84.3%). However, when using WorldPop top–down unconstrained or GPWv4, coverage was considerably lower at 51.5% and 44.7% respectively. This means that our observed differences could lead to very different conclusions when considering thresholds for accessibility coverages, such as those used in EPMM. Supplementary Fig. 55 shows the subnational discrepancies in accessibility coverage at a 2-hour travel

time threshold at administrative level 1. Although the differences are smaller than those in Fig. 4 (1-hour travel time), the unconstrained datasets show markedly different patterns than the constrained datasets.

In light of previous research and policy documents, that have relied on a single gridded population dataset for coverage estimates, our results also provide interesting clues for comparison across various population input data used in the same region or country. For example, studies on geographic access to care in Mozambique have used GPW[29], WorldPop top–down unconstrained[64], and HRSL[30], leading to different estimates. In addition, studies that have examined health service accessibility at a global or continental level, such as Weiss et al.[36] and Wigley et al.[65] have reported national and subnational accessibility coverages that fall outside the coverage ranges we found. For example, Weiss et al.[36] found an accessibility coverage of 78.7% in a 1-hour catchment in Madagascar, while our coverage estimates in the same catchment ranged from 58.7 to 76.6%. The same holds true for other countries where estimates of Weiss et al.[36] were either outside our range of estimates or we found large ranges around the reported estimate. Moreover, Wigley et al.[65] estimated that all countries in sub-Saharan Africa meet the target of 80% of pregnancies falling within a 2-hour catchment of the nearest hospital. However, our analysis suggests that at least 13 countries do not meet this target according to one gridded population dataset and 7 countries have either two or more datasets that indicate a coverage lower than 80%. While any comparison of coverage is also influenced by other input data used in an accessibility analysis, such as travel scenarios, road networks, and health facility coordinates, our results can be used to get a sense of the potential uncertainty in estimating coverage based on the population denominator chosen.

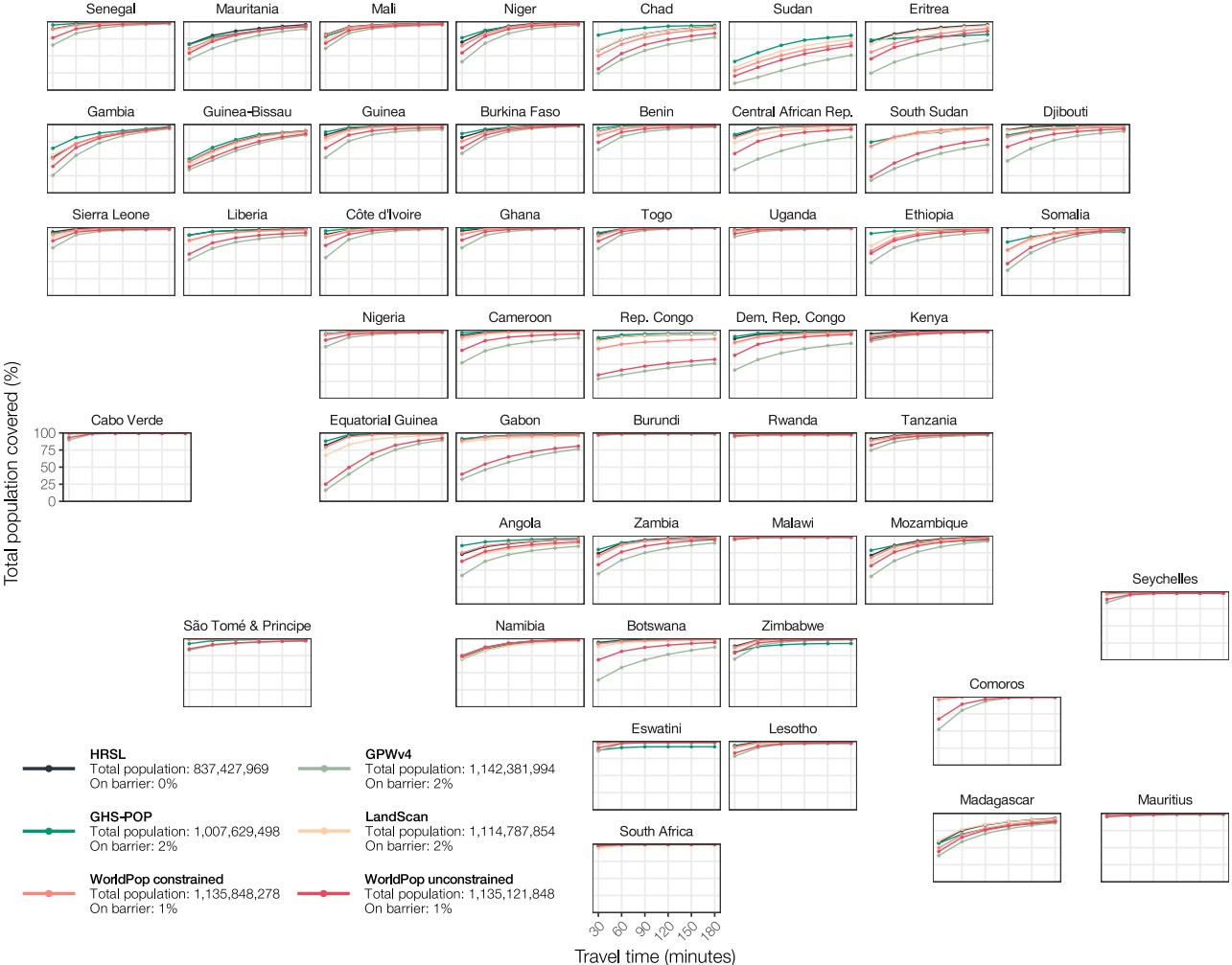

**Fig. 3 Accessibility coverage for all sub-Saharan African countries.** National plots for all sub-Saharan African countries comparing relative accessibility coverage statistics at 30, 60, 90, 120, 150, and 180 min travel time for the six gridded population data sets. Each plot corresponds to the relative geographical location of the country. Legend indicates the total population falling on barriers (i.e., permanent waterbodies) and thus not included in the analysis.

Gridded population datasets heavily rely on the recency and quality of population censuses, which countries commonly conduct every 10 years, however many countries in sub-Saharan Africa have not conducted a full population census in more than 15 years due to financial constraints, political instability, or remoteness[19,66]. In the Democratic Republic of the Congo, the last complete census was carried out in 1984, and policy-makers and gridded population data providers, therefore, rely on imprecise estimates of the current population through linear population projections[67]. The growing number of bottom-up population estimation approaches overcomes this challenge by conducting micro-censuses in small areas which are then extrapolated to larger administrative units using ancillary satellite data[56]. In January 2022, WorldPop released bottom-up population estimates for seven provinces (i.e., Haut-Katanga, Haut-Lomami, Ituri, Kasaï, Kasaï Oriental, Lomami, and Sud-Kivu) in the Democratic Republic of the Congo[68,69]. Interestingly, comparing the relative coverage estimates of the bottom-up and top–down datasets in these seven provinces did not lead to different patterns than earlier observed (Supplementary Table 4), meaning that the relative bottom-up coverage fell within the range of the constrained top–down datasets. However, absolute comparisons were markedly different, with generally lower total population counts in the bottom-up dataset and thus

proportionately lower numbers of people falling within the 1-hour health facility catchment. Even though it is impossible to indicate the best-gridded population dataset, objective comparisons of population products can improve our understanding of the differences and the implications of using one dataset in particular[20].

In terms of fitness for use, population datasets that constrain population to settled areas, based on high-resolution settlement data (i.e., HRSL, WorldPop top–down constrained), are more suited for accessibility modeling assuming acceptable levels of accuracy[57]. Most accessibility models need to consider the population at their place of residence (i.e., de jure/de facto population)[20], because the aim is to capture the complexity of the patient's journey to reach a health facility from their home, so that health system improvements can be targeted, and micro-planning of outreach is possible[54,55]. This is complicated when datasets do not constrain population to buildings or when ambient population is modeled and thus make GPWv4, World-Pop top–down unconstrained, and LandScan less favorable. However, the interpretation of built-up areas from satellite imagery is not without error. This means that in the absence of complete settlement data, these unconstrained datasets are still important and useful in ensuring that no population is overlooked in health estimates. Resolution and recency are other

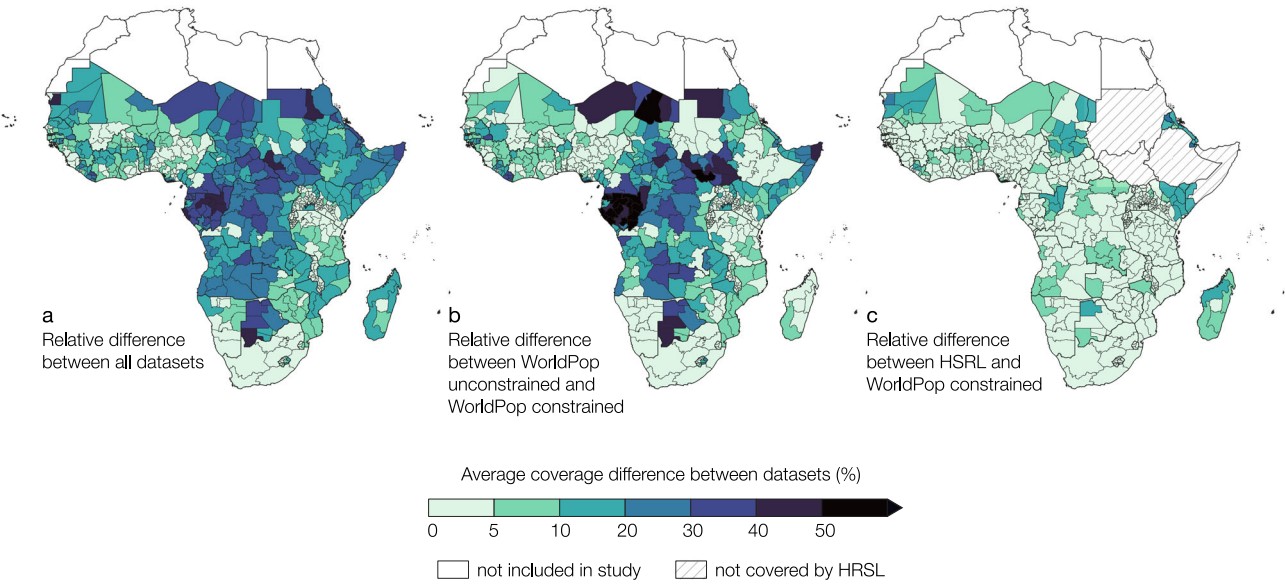

**Fig. 4 Subnational accessibility coverage maps for sub-Saharan Africa.** Relative accessibility coverage at a 1-hour travel time limit for **a** WorldPop top-down constrained, **b** WorldPop top-down unconstrained, **c** GPWv4, **d** LandScan, **e** GHS-POP, and **f** HRSL. Boundaries reflect administrative level 1.

**Fig. 5 Maps of relative difference in accessibility coverage estimates.** Maps show the average relative difference in accessibility coverage statistics at 1-hour travel time between **a** all datasets, **b** WorldPop top-down unconstrained and WorldPop top-down constrained, and **c** HSRL and WorldPop top-down constrained for full sub-Saharan Africa.

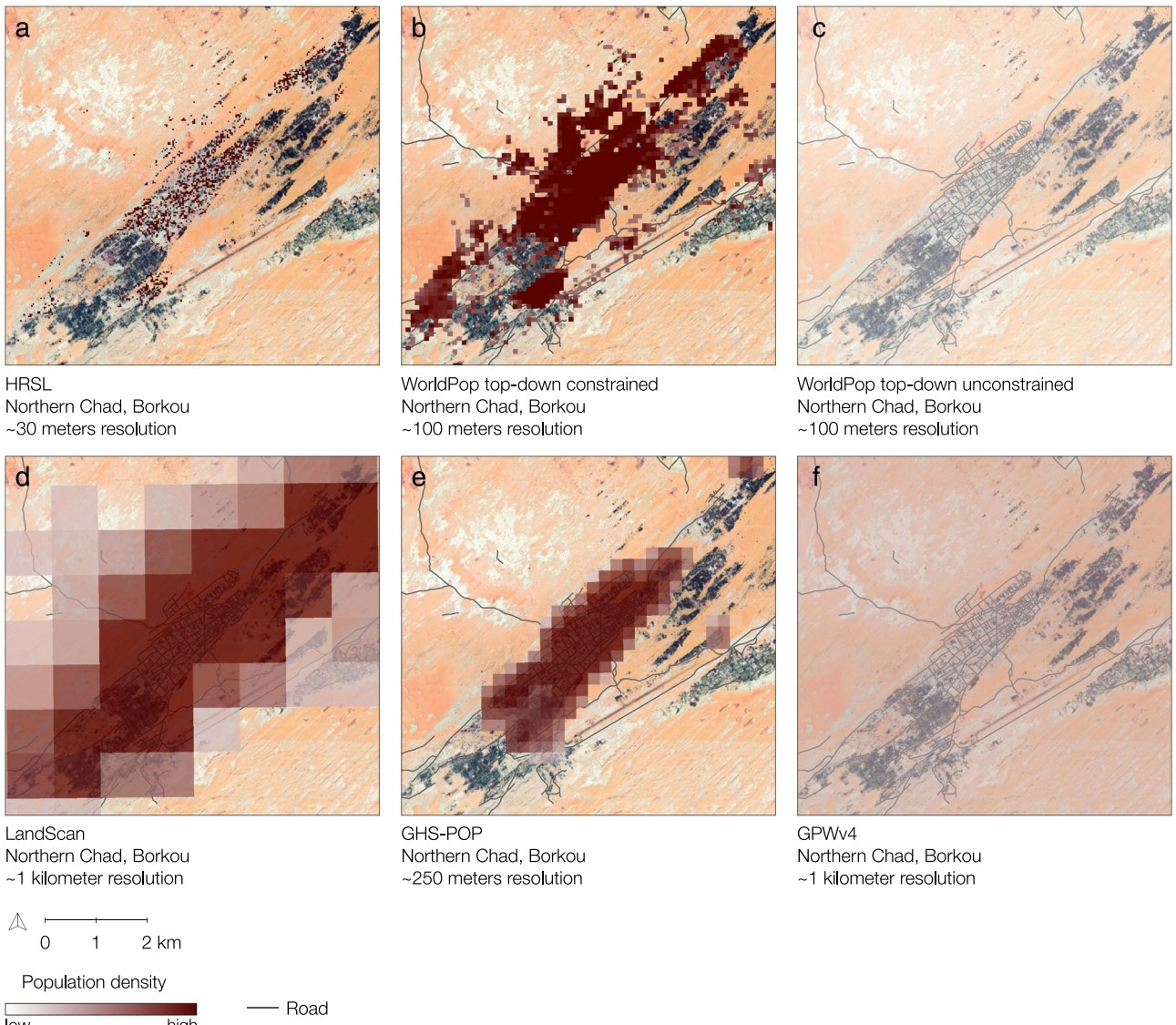

**Fig. 6 Visual comparison of gridded population datasets.** Visual differences between **a** HRSL, **b** WorldPop top–down constrained, **c** WorldPop top–down unconstrained, **d** LandScan, **e** GHS-POP, and **f** GPWv4 for Borkou, a northern region in Chad. Google satellite imagery as background (Map data© 2015 Google). White transparent color represents low numbers of population density.

important factors that weigh in this decision. Accessibility models are dependent on a more local target scale for analysis, the unconstrained population datasets cannot provide this analytical scale because the population is distributed over larger units. Constrained models focus more explicitly on the units of interest relevant for accessibility modeling and therefore the estimates are more plausible. The HRSL and WorldPop top–down constrained datasets seem the best fit for use when it comes to accessibility modeling to healthcare. Our advice would be to, where possible, consider both datasets and construct a plausible range of coverage estimates, comprising the mean, and a lower and upper bound around the summary statistics.

There are a number of caveats related to our accessibility analysis which are explained in more detail in the Methods. In brief, our travel time raster was developed while making large-scale assumptions on the speeds- and modes of transport that may not appropriately reflect local contexts. We think that by keeping the speeds and modes constant per country, we were able to more concretely discuss the differences in coverage as a result of the population datasets. Since this study aims at assessing the

uncertainty in accessibility coverage estimates associated with different population data, we simplified the assumptions involved in the definition of the travel time surface. For instance, we did not add lower and upper bounds to our travel speeds nor did we assume cross-border travel to seek healthcare. However, we acknowledge that different assumptions in other input data, such as travel speeds, health-seeking behavior, road networks, and barriers, may introduce further uncertainty into our coverage estimates. While global travel time surfaces have proven to be powerful and useful tools for advocacy[36], we contend that for local decision-making processes, it is essential to closely colla-borate with and consult local experts to produce realistic model outputs. As a consequence, the estimated accessibility statistics should not be used for local policy- or decision-making because they lack essential information needed to produce a realistic accessibility model.

In addition, we did not reallocate population falling on barriers (i.e., rivers and lakes). Across sub-Saharan Africa, this percentage is only 0–2% of the total population, but at the national and subna-tional level, these figures can be higher, especially in small island

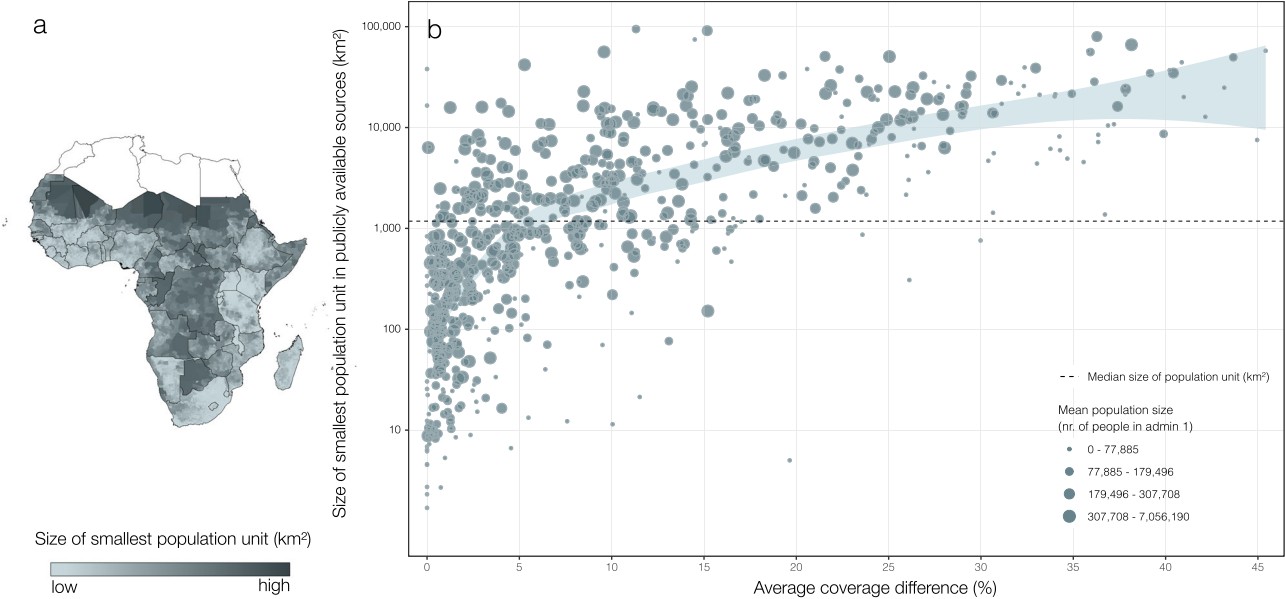

**Fig. 7 Association between size of population unit and difference in accessibility coverage. a** Spatial variation of the size of the smallest publicly available population unit. **b** Association between population input unit size and the average relative difference in accessibility coverage between the datasets. The size of the points indicates the total population averaged for all datasets at administrative level 1.

countries. However, since the observed percentages are generally low, the observed patterns are not expected to change dramatically. Furthermore, the version of the HRSL dataset used in this study did not include all the countries included in the analysis. In particular, Ethiopia, Somalia, South Sudan, and Sudan were not available for the continental HRSL raster, which affected comparisons between all datasets in these countries and resulted in widely diverging total population counts at the sub-Saharan African scale.

Despite an increasing global availability of data on numerous geographic objects, we still face challenges in precisely and accurately locating population, especially in low- and middle-income countries and areas sparsely populated. Yet, data on population density and distribution are vital inputs for research and policy-making[70]. The results presented in this study show how valuable and critical a comparative analysis between population datasets is for the derivation of coverage statistics that inform local policies and monitor global targets. Our results show that large differences exist between the datasets. This is also true for datasets informed by building footprints, even though at a smaller extent. Caution should be taken when drawing conclusions from any single gridded population dataset and potential uncertainties and limitations should increasingly be acknowledged in accessibility studies. A critical comparison of the results provided here shines a light on the sensitivity, reliability, and plausibility of coverage statistics.

## Data availability

The open and raw data that were used to derive the findings of this study are available via the relevant resources as indicated in Table 1 and Supplementary Table 2. The gridded population data from LandScan are not publicly available, but can be requested from their database, however, specific licensing rules may apply. All results that were obtained from this study are accessible in Supplementary Data 2. Source data for the main figures are available in Table 2 and Supplementary Data 2.

## Code availability

The R and Python code for data processing and analysis are available at Github [https://github.com/fleurhierink/Population_Access] and Zenodo [https://doi.org/10.5281/zenodo.7004009][45].

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

## Acknowledgements

P.M.M. is supported by the Royal Society Newton International Fellowship (NIF/R1/201418) and is also grateful for the support of the Wellcome Trust to the Kenya Major Overseas Program (# 203077). M.L. acknowledges support from Bill & Melinda Gates Foundation Grant number INV-009579. This research benefited from support through the National Science Foundation (award #1924670), and support provided to the University of Colorado Population Center (CUPC, Project 2P2CHD066613-06) from the Eunice Kennedy Shriver Institute of Child Health Human and Human Development. The content is solely the responsibility of the authors and does not necessarily represent the official views of NIH or CUPC.

## Author contributions

F.H. led the study along with N.R. Conceptualization of the study was done by F.H., N.R, and G.B. The methodology was initially developed by F.H., N.R., and G.B., and reviewed by all co-authors. Data analysis and processing were done by F.H. and supported by P.T., G.B., and N.R. Writing of the original draft was done by F.H. and supported by N.R. and G.B. Initial reviews on the figures were given by P.M.M., P.O., G.B., A.J.T., and N.R. The manuscript was edited by F.H., G.B., P.M.M., P.O., P.T., K.T., N.O., M.L., S.L., A.J.T., and N.R. All authors have further assisted in thoroughly reviewing all figures and texts.

## Competing interests

Peter Macharia is an Editorial Board Member for Communications Medicine, but was not involved in the editorial review or peer review, nor in the decision to publish this article. The authors have no competing interests to declare.
