## [Peer Review File · Communications Medicine]

Reviewers' comments:

Reviewer #1 (Remarks to the Author):

This is an important analysis and a well written paper! Undoubtedly this will provide important insights for population grid selection for countless future works.

Overall comments:

Limitations need to be more explicit and in the discussion. I felt like they were scattered throughout the paper, and many things just mentioned once (e.g. the fact that HRSL doesn't have Ethiopia, Somalia, South Sudan or Sudan data shouldn't only be mentioned as a footnote to figure 1)

Need more explicit figures for the examples you provide, namely Republic of Congo which is used as an exemplar many times.

Why use a newly created travel surface rather than using an existing one like Weiss et al?

Should have consistent use of oxford comma or not.

Specific comments by manuscript line:

Page 2, Line 3: Please clarify the among others here. What are the others?

Page 2, Lines 8 - 10 : This sentence is confusing as written

Page 2, Line 10: "Significant" - please only use this if a statistical test was done, otherwise use something like "most notable"

Page 2, Line 18: Please explain how geographic access is fundamental to organization of a health system. It is not clear from the current sentence how this may influence the health system.

Page 2, Line 22: Please see the following paper [1] which created an example of the placement of new facilities using similar methods

Page 3, Lines 33-34: the second half of this sentence is confusing as written

Page 3, Line 41/42: What are "Earth observations"?

Page 4, Line 57-59: This is a comparator sentence but doesn't have the comparison. I.e. The focus on general data properties compared to what?

Page 4, Lines 64-68: Can you define the methodology for these somewhere briefly, maybe in the supplement?

Page 5, Line 80: Can you clarify if these are positive or negative changes?

Page 5, Lines 91-98: This should really be a table to visually see the % covered by different travel

times globally.

Page 6: This is where I think we really need a figure of Congo to reference.

Page 8: Same comment here - these really need specific figures to reference

Page 10, Lines 216-221: This is really more of discussion

Page 12, lines 269-276: I'm struggling to understand how the comparison to one specific paper, while not even using the same input data. I think the points made are helpful but could be made more generally rather than specifically to the paper by Weiss et al. especially considering that they had their own travel layer and used somewhat differing facility lists.

Page 15: There are several limitations missing. The HRSL limitation needs to be mentioned here, as does a lack of uncertainty around any modeled estimates, and the travel speed assumptions made. Similarly, need to mention that each country was modeled separately, which is a limitation for regional interpretation

Page 16, Line 351: Explain why isotropic was chosen / why no corrections were made for slopes.

Page 16, Lines 353-361: This is a limitation to me. Recent analyses [1] have shown that for several countries and populations, the nearest facility is cross border. This is unaccounted for in your analysis. How can you address this?

Page 17, Line 385: Does this include pharmacies?

Page 19, Line 435: This needs to be part of the discussions

Figure 1: The HRSL constraint needs to be discussed earlier

Figure 3: This is really hard to see - can it be made bigger?

[1] Hulland, E.N., Wiens, K.E., Shirude, S. et al. Travel time to health facilities in areas of outbreak potential: maps for guiding local preparedness and response. *BMC Med* 17, 232 (2019).
<https://doi.org/10.1186/s12916-019-1459-6>

Reviewer #2 (Remarks to the Author):

This study explores the uncertainty introduced in assessing health care accessibility in sub-Saharan Africa due to the use of different population estimates. Modelling population estimates at high spatial resolution is important for areas where such information is not routinely collected. Understanding the uncertainties introduced in spatial analyses based on these modelled estimates is important to assess the accuracy of outputs.

The study is very well written and overall sound. I only have some minor comments:

1. Introduction: Page 3: I suggest moving the aim of the study to the end of the introduction
2. Results: some of the subheadings such as 'Sub Saharan Africa', 'National' etc could be expanded on to make them more meaningful

3. Page 6: It is not fully clear what the authors mean by 'lower coverage statistics', could this be better specified throughout the paragraph
4. Page 7/10: What are administrative levels 1 and 2? Could the authors clarify this to aid understanding of the context
5. Page 26: Making code available via a GitHub page or similar would allow for wider accessibility of the developed methods. Asking for code from authors is often a barrier

Reviewers' comments:

Reviewer #1 (Remarks to the Author):

This is an important analysis and a well written paper! Undoubtedly this will provide important insights for population grid selection for countless future works.

We thank the reviewer for the thoughtful feedback on our manuscript. We are particularly grateful for the comments made on the limitations of our study and on the construction of the travel time surfaces. We have provided a point-by-point response to the reviewer's comments below and have addressed them in the manuscript. We hope that our changes are to the reviewer's satisfaction.

Overall comments:

Limitations need to be more explicit and in the discussion. I felt like they were scattered throughout the paper, and many things just mentioned once (e.g. the fact that HRSL doesn't have Ethiopia, Somalia, South Sudan or Sudan data shouldn't only be mentioned as a footnote to figure 1)

We fully agree with the reviewer — the limitations of our study should be better organized and presented in the discussion section. We have added a statement on the limitations of the HRSL raster at the beginning of the result section. We have also further elaborated the limitations in the discussion section.

Need more explicit figures for the examples you provide, namely Republic of Congo which is used as an exemplar many times.

We have created additional figures for the examples introduced in the results and discussion sections and for all other countries presented in this study. These figures show the relative accessibility coverage at 30, 60, 90, 120, 150, and 180 minutes at administrative level 1. Following the journal guidelines, we have included these figures as a Supplementary File.

Why use a newly created travel surface rather than using an existing one like Weiss et al?

Thank you very much for your critical reflection on this point. The Weiss et al. travel surface indeed provides a globally available friction raster that can be adapted to other contexts. However, the friction surface as created by Weiss et al. is available at a resolution of 1km x 1km, which for our purpose of use was too coarse to realistically represent barriers. In addition, Weiss et al. also used waterbodies under the assumption that they could be traversed at a swimming speed of 1km an hour. Since we only used water polygons and lines which were classified as generally larger (e.g. lakes and rivers. Streams were removed), non-traversable features, we wanted to treat them as full barriers to the population. Furthermore, most of the input data we used was available at approximately 100 meters resolution. All gridded population data were available within the range of 30 meters to 1 kilometers resolution so creating a travel time surface at 100 meters resolution seemed to be the right compromise between spatial accuracy and computational efficiency. In addition, Weiss et al. used global default travel speeds for road classes which did not have a speed limit nor average speed information. In our paper we decided to calculate the average speed per road class on the African continent, which was considered to be a better fit for the context of sub-Saharan Africa.

Most importantly, the travel time surfaces created in this paper were used as an indicator to assess the impact of the differences between gridded population products and not to

inform the research community on coverage statistics. To inform policy- and decision-making processes it is preferred to work at smaller scale to ensure higher accuracy and more realism in the model inputs by consulting local experts on health facility data and information of health seeking behavior of the target population, so that the travel scenarios can be best adapted to the local context.

*We have added a section at the end of the “**accessibility modelling**” sub-section in the methods section to better describe these considerations. We hope this sufficiently answers the question raised by the reviewer.*

Should have consistent use of oxford comma or not.

We indeed found some instances where the oxford comma should have been used. We have now included them in the manuscript.

Specific comments by manuscript line:

We thank the reviewer for the thorough review of the paper and have made changes throughout the manuscript, which are described in more detail below.

Page 2, Line 3: Please clarify the among others here. What are the others?

We have clarified this sentence by adding: “Geographic access to healthcare can be modelled using spatial datasets on terrain, together with the distribution of existing health facilities and populations.”

Page 2, Lines 8 - 10: This sentence is confusing as written

We have rephrased the sentence as follows: “The differences are most notable in large and sparsely populated administrative units and dramatically shape patterns of health care accessibility at national and sub-national levels.”

Page 2, Line 10: “Significant” – please only use this if a statistical test was done, otherwise use something like “most notable”

We have replaced “significant” with “notable.”

Page 2, Line 18: Please explain how geographic access is fundamental to organization of a health system. It is not clear from the current sentence how this may influence the health system.

We have reworded the sentence as follows: “... it is fundamental to the organization of a health system as it determines the spatial reach of health services in relation to the population.”

Page 2, Line 22: Please see the following paper [1] which created an example of the placement of new facilities using similar methods

We thank the reviewer for the suggestion. We have added the reference in the manuscript.

Page 3, Lines 33-34: the second half of this sentence is confusing as written

We have changed the sentence as follows: “It is not known to what extent the use of different population data in accessibility analyses affects accessibility coverage (i.e., the proportion of the population that can access a health facility within a given travel time threshold) and thus monitoring indicators that underpin policy-making at the global, national, and sub-national level.”

Page 3, Line 41/42: What are "Earth observations"?

We have changed the sentence as follows: “Gridded population datasets allocate population counts across rows and columns of grid cells either by using simple techniques

to uniformly redistribute census data or by using ancillary variables coming from Earth observations (e.g., land cover, elevation, night lights, etc.), population data, and dasymetric modelling techniques, that provide more refined estimates¹⁹.”

Page 4, Line 57-59: This is a comparator sentence but doesn't have the comparison. I.e. The focus on general data properties compared to what?

We have changed the sentence as follows: “However, the focus is often on general data characteristics and their suitability^{19,20,31} or on country- or discipline-specific implications of using the different data products^{21,32-34}, rather than quantifying differences in model outcomes at large geographical scales”.

Page 4, Lines 64-68: Can you define the methodology for these somewhere briefly, maybe in the supplement?

We agree with the reviewer that some background information on the methods may help to better understand the differences between datasets. We believe that the column “Method” in Supplementary Table 1 captures the main principles of the reallocation techniques used to produce the different datasets. We also have included references to research focusing on the methodological differences between these datasets, such as:

- *Leyk Stefan, Gaughan Andrea E, Adamo Susana B, et al. The spatial allocation of population: A review of large-scale gridded population data products and their fitness for use. Earth System Science Data 2019;11(3)*
- *Thematic Research Network on Data and Statistics (TReNDS). Leaving no one of the map: A guide for gridded population data for sustainable development. 2020. [Available from]: <https://static1.squarespace.com/static/5b4f63e14eddec374f416232/t/5eb2b65ec575060f0adb1feb/1588770424043/Leaving+no+one+off+the+map-4.pdf>*
- *Yin Xu, Li Peng, Feng Zhiming, et al. Which Gridded Population Data Product Is Better? Evidences from Mainland Southeast Asia (MSEA). ISPRS International Journal of Geo-Information 2021;10(10):681.*

Page 5, Line 80: Can you clarify if these are positive or negative changes?

We have changed the sentence as follows: “Datasets that distribute population over larger land areas, rather than being limited to building footprints, result in longer travel times for a portion of the population and therefore lower overall estimates of accessibility, drastically changing accessibility patterns.”

Page 5, Lines 91-98: This should really be a table to visually see the % covered by different travel times globally.

We agree with the reviewer and have added Table 1 that reflects these numbers.

Page 6: This is where I think we really need a figure of Congo to reference.

Like a map or a table for the supplementary?

We fully agree with the reviewer. However, following the journal guidelines on the maximum number of figures, we have created an additional Supplementary File with relative coverage statistics for the 30, 60, 90, 120, 150, and 180 minutes travel time catchments at administrative level 1 for all countries included in this study.

Page 8: Same comment here - these really need specific figures to reference

Like a map or a table for the supplementary?

We fully agree with the reviewer. Following the logic presented above, we have added these figures in the Supplementary File.

Page 10, Lines 216-221: This is really more of discussion

We understand that this section could potentially fit well in the discussion. However, we contend that the paragraph describes the results presented in Figure 7 in a compelling fashion. It condenses the wealth of information that could be extracted from all figures and starts the transition from the results into the discussion. Although we greatly appreciate the comment made by the reviewer, we decided not to alter the sentence's original location.

Page 12, lines 269-276: I'm struggling to understand how the comparison to one specific paper, while not even using the same input data. I think the points made are helpful but could be made more generally rather than specifically to the paper by Weiss et al. especially considering that they had their own travel layer and used somewhat differing facility lists.

We agree with the reviewer that the comparison should be more general and also involve other examples. We have completed the paragraph by discussing the paper of Wigley et al., where the last sentence of the paragraph indicates that these coverage statistics are indeed dependent on other input data, but this study can give a sense of the uncertainty introduced by the population data.

Page 15: There are several limitations missing. The HRSL limitation needs to be mentioned here, as does a lack of uncertainty around any modeled estimates, and the travel speed assumptions made. Similarly, need to mention that each country was modeled separately, which is a limitation for regional interpretation.

We thank the reviewer for the insightful comment. We have completed the Discussion section by adding a sentence discussing the limitations highlighted.

Page 16, Line 351: Explain why isotropic was chosen / why no corrections were made for slopes.

We used an isotropic analysis to simplify model assumptions. The primary goal of this study was to quantify the magnitude of differences in accessibility coverage estimates rather than constructing a new continental travel time surface. We acknowledge that for the construction of a reliable and useful travel time surface, we should consider additional information on local context, particularly on country-specific health seeking behaviors. While global travel time surfaces have proven to be useful tools for advocacy, we contend that for local decision-making processes one should always closely collaborate with local experts to ensure realistic model outputs. In this specific study slope corrections were more difficult to apply because we did not have information on modes of transport for the different road types. In addition, for some OSM road classes, we did not have accurate speed information, which complicated assumptions on modes of transport. Since slope corrections are usually made for walking and biking speeds having local information on modes of transport is crucial and this will likely differ between countries and regions of a country.

*We have added a couple of sentences to the "**Accessibility model**" section in the Methods to address these considerations.*

Page 16, Lines 353-361: This is a limitation to me. Recent analyses [1] have shown that for several countries and populations, the nearest facility is cross border. This is unaccounted for in your analysis. How can you address this?

We agree with the reviewer that health seeking behavior is a dynamic and complex dimension of accessibility analyses and that having information on local context is key in developing a travel time surface that can be used as decision-making tool. However, we do not recommend using our sub-Saharan travel time raster as a decision-making tool due to the relatively simplistic assumptions made on travel modes and speeds, barriers to the

population, and other health seeking specificities. The aim of this paper was to compare the impact of using different gridded population data in accessibility models. In order to simplify this comparison and not bias our model by including other different or hard-to-capture aspects of health seeking specificities, we did not allow for cross-border health seeking, as this would further complicate the comparison of accessibility statistics. To do so, we would need to know which part of the population seeks health care outside the country, which would complicate the comparison of accessibility statistics at national and subnational levels. We have added a sentence to the discussion to indicate that we did not assume cross-border health seeking.

Page 17, Line 385: Does this include pharmacies?

The dataset as produced by Maina et al. (2019) only captures public health facilities. Pharmacies are not a part of the categories of health facilities in the dataset.

Page 19, Line 435: This needs to be part of the discussions

We agree with the reviewer and moved the sentence to the Discussion section, which now reads: "In addition, we did not reallocate population falling on barriers (i.e., rivers and lakes). Across sub-Saharan Africa, this percentage is only 0-2% of the total population, but at national and sub-national levels these figures are higher, especially in small island states. Since the percentages are low overall, this is not expected to change the patterns we have observed."

Figure 1: The HRSL constraint needs to be discussed earlier

We have added a sentence to the first paragraph of the results, which now reads: "Importantly, HRSL data for Ethiopia, Somalia, South Sudan, and Sudan were not available at the time of this study."

Figure 3: This is really hard to see - can it be made bigger?

We have increased the size of the axis labels and will ensure that upon final submission all our figures will be uploaded in pdf format to ensure vectorized quality and allow zooming for the online readers. Figure 3 provides an overarching overview of country-specific trends, which impedes to further enlarge the individual plots. To overcome this limitation, we have created an additional Supplementary File for all countries in the study area (n = 49) that provide the coverage statistics at 30, 60, 90, 120, 150, and 180 minutes travel time at the administrative level 1. These figures also improve the comprehension of the examples given for Chad, Sudan, Eritrea, South Sudan, Central African Republic, Republic of the Congo, Democratic Republic of the Congo, Equatorial Guinea, and Gabon.

[1] Hulland, E.N., Wiens, K.E., Shirude, S. et al. Travel time to health facilities in areas of outbreak potential: maps for guiding local preparedness and response. BMC Med 17, 232 (2019). <https://doi.org/10.1186/s12916-019-1459-6>

Reviewer #2 (Remarks to the Author):

This study explores the uncertainty introduced in assessing health care accessibility in sub-Saharan Africa due to the use of different population estimates. Modelling population estimates at high spatial resolution is important for areas where such information is not routinely collected. Understanding the uncertainties introduced in spatial analyses based on these modelled estimates is important to assess the accuracy of outputs.

We thank the reviewer for the thorough and positive feedback on our manuscript. We have carefully considered all the comments of the reviewer and made changes throughout the manuscript accordingly and in the point-by-point response below. We hope that the revisions are to the reviewer's satisfaction.

The study is very well written and overall sound. I only have some minor comments:

1. Introduction: Page 3: I suggest moving the aim of the study to the end of the introduction

We thank the reviewer for the thoughtful reflection. Even though we agree with the reviewer that moving the aim to the end of the introduction could be logical, we decided to leave the sentence where it is now because we feel that it supports the reader in comprehending the first part of the introduction which is quite dense in information. The aim is repeated in a more elaborated and complete manner at the end of the introduction, which gives more insights into the nature of the analysis. We hope that we have sufficiently addressed the reviewer's comment.

2. Results: some of the subheadings such as 'Sub Saharan Africa', 'National' etc could be expanded on to make them more meaningful

We agree with the reviewer and have changed the sub-headings into more self-explanatory titles (see list below):

- *Diverging accessibility coverage estimates for sub-Saharan Africa*
- *Comparing national differences in accessibility coverage allows localization of high variation*
- *Differences in accessibility coverage are most evident at sub-national scale*

3. Page 6: It is not fully clear what the authors mean by 'lower coverage statistics', could this be better specified throughout the paragraph

We have changed the first sentence that includes the term "lower coverage statistics" to: "In these countries, we observe lower coverage statistics for GPWv4 and WorldPop top-down unconstrained, meaning that a relatively smaller number of people have access to healthcare, sometimes followed by significant discrepancies between coverage values for the other datasets." We believe that this change improves the readability throughout the paragraph.

4. Page 7/10: What are administrative levels 1 and 2? Could the authors clarify this to aid understanding of the context

We have added a sentence to the first paragraph of this section, as follows: "While the naming of administrative levels varies from country to country, administrative level 0 always represents the national borders of a country, administrative level 1 represents the largest subnational unit of a country, smaller subnational levels are levels 2, 3 and 4."

5. Page 26: Making code available via a GitHub page or similar would allow for wider accessibility of the developed methods. Asking for code from authors is often a barrier

We fully agree with the reviewer and created an openly accessible Github page. All the code is now available on: https://github.com/fleurhierink/Population_Access

REVIEWERS' COMMENTS:

Reviewer #1 (Remarks to the Author):

I thank the authors for the extensive revisions. I find the replies and changes to the manuscript satisfactory.

I did notice one small issue with Table 1 versus the manuscript and supplements - the naming of the 'Facebook' dataset tends to be inconsistent. In Table 1 it is 'Facebook' but in Supplement Table 1 and in the manuscript it is often referred to as HRSL. Please use consistent language.

Aside from the above minor discrepancy, I feel like all my concerns and questions have been answered.

Reviewer #2 (Remarks to the Author):

The authors have fully addressed my comments and I do not have any further comments.

REVIEWERS' COMMENTS:

Reviewer #1 (Remarks to the Author):

I thank the authors for the extensive revisions. I find the replies and changes to the manuscript satisfactory.

I did notice one small issue with Table 1 versus the manuscript and supplements - the naming of the 'Facebook' dataset tends to be inconsistent. In Table 1 it is 'Facebook' but in Supplement Table 1 and in the manuscript it is often referred to as HRSL. Please use consistent language.

Aside from the above minor discrepancy, I feel like all my concerns and questions have been answered.

We thank the reviewer for the time and effort invested in reviewing this manuscript. The suggestions and comments in an earlier round of review were helpful and aided the authors to further improve the manuscript.

We have adjusted the tables where we referred to the HRSL dataset as Facebook to match the other figures and tables.

Reviewer #2 (Remarks to the Author):

The authors have fully addressed my comments and I do not have any further comments.

We thank the reviewer for his thorough reviews now and in an earlier round, which have supported some additional critical thinking on the paper and helped the completion of this manuscript.